# A decoherence explanation of the gallium neutrino anomaly

Yasaman Farzan[1*] and Thomas Schwetz[2†]

**1** School of physics, Institute for Research in Fundamental Sciences (IPM),
P.O.Box 19395-5531, Tehran, Iran
**2** Institut für Astroteilchenphysik, Karlsruher Institut für Technologie (KIT),
76021 Karlsruhe, Germany

★ yasaman@theory.ipm.ac.ir , † schwetz@kit.edu

## Abstract

Gallium radioactive source experiments have reported a neutrino-induced event rate about 20% lower than expected with a high statistical significance. We present an explanation of this observation assuming quantum decoherence of the neutrinos in the gallium detectors at a scale of 2 m. This explanation is consistent with global data on neutrino oscillations, including solar neutrinos, if decoherence effects decrease quickly with energy, for instance with a power law $E_\nu^{-r}$ with $r \simeq 12$. Our proposal does not require the presence of sterile neutrinos but implies a modification of the standard quantum mechanical evolution equations for active neutrinos.

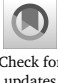
# 1   Introduction

The gallium solar-neutrino detectors GALLEX [1,2] and SAGE [3,4] have been used to measure the neutrino induced event rate from radioactive $^{51}$Cr and $^{37}$Ar sources, leading to event rates consistently lower than Standard Model (SM) expectations. This has been called the *gallium anomaly* [5–7]. Recently, the BEST collaboration has performed a dedicated source experiment using a $^{51}$Cr source in the center of a two-volume gallium detector, confirming the previous hints by observing an event deficit of around 20% compared to the SM prediction at high statistical significance [8,9].

Traditionally, the gallium anomaly has been interpreted in terms of sterile neutrino oscillations, see e.g. [7,10]. This explanation, however, is in sever tension with constraints from solar neutrinos as well as short-baseline reactor experiments [11–14]. It is unlikely that the anomaly can be resolved by cross section uncertainties [12,15,16]. The authors of ref. [17] discuss various possible explanations based on conventional or non-standard physics (see also [18]), with no convincing solution. Therefore, at present the gallium anomaly remains a puzzle.

Below we are going to present an explanation of the gallium anomaly in terms of quantum-decoherence of neutrinos. We are assuming a modification of standard quantum mechanics by some exotic new physics which induces a loss of coherence in the evolution of the quantum states [19–21]; for previous applications to neutrino oscillations see e.g., [22–33]. We postulate that in the evolution of the electron neutrinos produced in the gallium experiments, decoherence is lost already at distances of order few meters, in order to explain the observed event deficit. In a similar spirit as in the "soft decoherence" scenario of [26] we assume that decoherence effects dominate at low energies and are suppressed with increasing energy by a steep power law. In this way we can explain the gallium anomaly without impacting other oscillation measurements. As we discuss, this scenario is also consistent with the solar neutrino data.

Hence, we offer an explanation of the gallium anomaly without affecting the success of the standard three-flavour explanation of oscillation data. To explain the LSND hint for $\bar{\nu}_\mu \to \bar{\nu}_e$ transitions [34], our explanation of the gallium anomaly can be combined with the decoherence model for LSND proposed in [28]. To simultaneously explain the LSND and gallium anomalies, we may allow for different decoherence parameters for neutrinos and antineutrinos or accept that decoherence effects happen only around neutrino energies of 0.75 and 30 MeV, but not in between or at higher energies. We do not address the MiniBooNE [35] anomaly, which requires an alternative explanation to this scenario. Similarly, our model predicts no non-standard effects at short-baseline reactor experiments (see [36] for a discussion in view of recent developments related to reactor neutrino flux predictions).

Our model is based on the decoherence of the three standard-model neutrinos and does not require an introduction of sterile neutrinos. Recent discussions of decoherence in oscillations of eV-scale sterile neutrinos can be found in refs. [37,38]. Let us stress that the decoherence that we postulate here requires exotic new physics which modifies the standard quantum mechanical evolution; conventional decoherence based on particle localisation leads only to tiny effects which are negligible for all oscillation experiments considered here [39,40].

The outline of the rest of the paper is as follows. In section 2, we introduce the quantum decoherence framework and identify the parameters of our scenario. Section 3 contains our considerations of the gallium anomaly: we present our numerical analysis and determine the decoherence parameters which can explain the gallium data. In section 4, we show that our scenario can be consistent with the global data on the neutrino oscillations, provided that the decoherence effects decrease rather quickly with energy in order to be compatible with the solar and reactor neutrino data. We comment on the possibility to also explain the LSND results along with the gallium data. We summarize our findings in section 5.

## 2 The decoherence model

In the decoherence model, the evolution of the density matrix, $\rho$ is modified as follows

$$\frac{d\rho}{dt} = -i[H,\rho] - \mathcal{D}[\rho].\tag{1}$$

While $H$ is the standard Hamilton operator, $\mathcal{D}$ accounts for the decoherence. To maintain complete positivity, $\mathcal{D}[\rho]$ has to be of Lindblad form [41,42]

$$\mathcal{D}[\rho] = \sum_n [\{\rho, D_n D_n^\dagger\} - 2D_n \rho D_n^\dagger].\tag{2}$$

To ensure unitarity, *i.e.*, $d\mathrm{Tr}(\rho)/dt = 0$, we impose the condition $D_n^\dagger = D_n$. This also guarantees the second law of the thermodynamics [42]. If we furthermore want the average energy $\mathrm{Tr}(\rho H)$ to be conserved, $H$ and $D_n$ should be simultaneously diagonalized: $[H, D_n] = 0$.

From now on, we take a single $D$ matrix. With the properties mentioned above, we can write the Hamiltonian and the $D$ matrix in the neutrino mass basis as

$$H = \frac{1}{2E_\nu}\mathrm{diag}(m_1^2, m_2^2, m_3^2), \qquad D = \mathrm{diag}(d_1, d_2, d_3),\tag{3}$$

where $m_i$ are the neutrino masses and $d_i$ are real quantities with dimension of square-root of mass. The decoherence terms lead to exponential damping of the off-diagonal elements of the density matrix, see e.g., [26], with a rate set by the decoherence parameters

$$\gamma_{ij} = (d_i - d_j)^2.\tag{4}$$

For instance, we obtain for the $\nu_e$ survival probability

$$P_{ee} = \sum_{i=1}^3 |U_{ei}|^4 + \sum_{i \neq j} |U_{ei}|^2 |U_{ej}|^2 e^{-\gamma_{ij} L} e^{-i\phi_{ij}},\tag{5}$$

where

$$\phi_{ij} = \frac{\Delta m_{ji}^2 L}{2E_\nu}.\tag{6}$$

Deviations from the standard oscillation formula are controlled by the decoherence parameters $\gamma_{ij}$. They have units of inverse length and an unknown energy dependence. Following the usual practice in the literature, we will assume here an arbitrary power law dependence for $\gamma_{ij}$ as

$$\gamma_{ij} = \frac{1}{\lambda_{ij}}\left(\frac{E_{\mathrm{ref}}}{E_\nu}\right)^r,\tag{7}$$

where $\lambda_{ij}$ is the decoherence length at a reference energy $E_{\mathrm{ref}}$, which we choose as $E_{\mathrm{ref}} = 0.75$ MeV, close to the dominant neutrino energies from a Cr source.

In the phenomenological study below, we will take $\lambda_{12}, \lambda_{13}$ and the power index $r$ as the independent parameters; $\lambda_{23}$ is then determined by using eq. (4), which implies that $\gamma_{23}$ is fixed up to a sign ambiguity:

$$\gamma_{23} = \gamma_{12} + \gamma_{13} \pm 2\sqrt{\gamma_{12}\gamma_{13}}.\tag{8}$$

Within the three active neutrino framework, for the gallium experiments the oscillation phases $\phi_{ij} \ll 1$ and we have $e^{i\phi_{ij}} \approx 1$.

Note that in eq. (3) we have assumed that matter effects are negligible and adopted the vacuum Hamiltonian. If matter effects are important, $H$ and $D$ will no longer commute. In such a case additional damping effects may appear, not only damping the off-diagonal elements of $\rho$, but also driving $\rho$ towards a matrix proportional to the identity matrix, see e.g., [25, 27, 30, 31]. We will come back to this in section 4.1, when discussing the solar neutrinos.

Table 1: Ratio $R$ of observed and predicted event numbers in the gallium experiments, assuming the two recomended cross sections CS1 and CS2 from Haxton et al. [16]. The quoted errors correspond to the $1\sigma$ combined statistical and uncorrelated systematic experimental uncertainties. The uncertainty on the cross section is not included.

|  | CS1 | CS2 |
|---|---|---|
| Gallex 1 [2] | $0.970 \pm 0.112$ | $0.946 \pm 0.109$ |
| Gallex 2 [2] | $0.826 \pm 0.102$ | $0.806 \pm 0.099$ |
| SAGE (Cr) [3] | $0.967 \pm 0.122$ | $0.944 \pm 0.119$ |
| SAGE (Ar) [4] | $0.805 \pm 0.085$ | $0.790 \pm 0.084$ |
| BEST (inner) [8] | $0.805 \pm 0.045$ | $0.786 \pm 0.044$ |
| BEST (outer) [8] | $0.779 \pm 0.046$ | $0.761 \pm 0.045$ |

## 3 Numerical analysis for gallium data

### 3.1 Discussion of the gallium anomaly

Gallium experiments consist of detector volumes with typical dimensions of few meters filled with gallium. In particular, the BEST experiments has two separated volumes, an inner spherical volume with radius 0.67 m and an outer cylindrical volume with radius 1.09 m and height 2.35 m [8,9]. In the center of these volumes they deploy intense radioactive sources providing a flux of $\nu_e$ from electron-capture decay. For the $^{51}$Cr source used in most measurements, the dominant neutrino energy lines are around 750 keV (430 keV) with branching ratios around 90% (10%). The $^{37}$Ar source used in one measurement campaign of the SAGE experiment has two dominant lines close to 812 keV, see e.g., [16] for more details.

The gallium source experiments report the ratio of observed to expected events

$$R = \frac{N^{\text{obs}}}{N^{\text{pred,SM}}}, \tag{9}$$

where $N^{\text{pred,SM}}$ is the number of events predicted in the SM without any $\nu_e$ disappearance, which requires to specify the cross section for the detection reaction $^{71}$Ga$(\nu_e, e^-)^{71}$Ge for neutrinos from $^{51}$Cr or $^{37}$Ar sources, correspondingly. There has been some discussion in the literature, to what level uncertainties on the cross section can affect the anomaly, e.g., [12,15–17]. In this work we adopt the recent detailed consideration of the relevant cross section from Haxton et al. [16]. They obtain corrections to the ground state transition leading to a ground state cross section about 2.5% smaller than Bahcall [43]. Then they provide two independent evaluations of the excited state transition, one based on $(p,n)$ measurements from Krofcheck et al. (1985) [44] and one using $(^3\text{He}, t)$ data from Frekers et al. (2011) [45]. In the following we denote the corresponding cross sections by (CS1) and (CS2), respectively. Including a detailed evaluation of the uncertainties, they obtain the following two "recommended" cross sections [16]

$$\begin{aligned} \sigma(^{51}\text{Cr}) &= 5.71^{+0.27}_{-0.10}, & \sigma(^{37}\text{Ar}) &= 6.88^{+0.34}_{-0.13}, & \text{(CS1)} \\ \sigma(^{51}\text{Cr}) &= 5.85^{+0.19}_{-0.13}, & \sigma(^{37}\text{Ar}) &= 7.01^{+0.22}_{-0.16}, & \text{(CS2)} \end{aligned} \tag{10}$$

in units of $10^{-45}$ cm$^2$ and quoted uncertainties at 68% CL.

In table 1 we report the ratios $R$ for the 6 data points, using either the (CS1) or (CS2) cross sections. The errors in the table include statistical and uncorrelated experimental systematic uncertainties. In order to evaluate the significance of the effect, they need to be combined with

Table 2: Evaluating the null-hypothesis $R = 1$ for the BEST experiments (inner and outer volumes combined) and for all gallium experiments, for the two recommended cross sections CS1 and CS2 from Haxton et al. [16]. We give the $\chi^2/$dof for the null-hypothesis and the corresponding $p$-values. In the bracket the $p$-values are converted into two-sided Gaussian standard deviations. The analysis includes experimental uncertainties as well as the cross section uncertainties as provided in [16].

|  | $\chi^2_{\text{null}}/$dof | $p$-value |
|---|---|---|
| CS1, BEST | 32.1/2 | $1.1 \times 10^{-7}$ (5.3$\sigma$) |
| CS1, all | 36.3/6 | $2.4 \times 10^{-6}$ (4.7$\sigma$) |
| CS2, BEST | 34.7/2 | $2.9 \times 10^{-8}$ (5.5$\sigma$) |
| CS2, all | 38.4/6 | $9.4 \times 10^{-7}$ (4.9$\sigma$) |

the correlated uncertainty due to the cross sections from eq. (10). To test the null-hypothesis of no neutrino disappearance we define

$$\chi^2_{\text{null}} = \min_{\xi_{\text{CS}}} \left[ \sum_i \frac{(1 + \delta^i_{\text{CS}} \xi_{\text{CS}} - R_i)^2}{\sigma_i^2} + \xi_{\text{CS}}^2 \right], \tag{11}$$

with $R_i$ and $\sigma_i$ given in table 1 and the index $i$ runs over the used data points; $\delta^i_{\text{CS}}$ is the relative uncertainty of the cross section derived from eq. (10), which depends on the index $i$ whether a Cr or Ar source has been used. In order to take into account the asymmetric cross section errors we use for $\delta^i_{\text{CS}}$ the upper (lower) error if the value of the pull parameter $\xi_{\text{CS}}$ at the minimum is larger (smaller) than zero. The results of this test are summarized in table 2, where we give the $\chi^2$ of the null-hypothesis for using only the two BEST data points or for combining all 6 gallium data points. We see that for both cross sections, very low $p$-values are obtained, corresponding roughly to 5$\sigma$ significance, with CS2 leading to slightly higher significances.

### 3.2 Fitting gallium data with the decoherence model

To test the decoherence model introduced in section 2, we modify the $\chi^2$ definition from eq. (11) in the following way:

$$\chi^2 = \min_{\xi_\alpha} \chi^2(\xi_\alpha), \qquad \alpha = \text{CS}, \theta_{12}, \theta_{13}, \tag{12}$$

$$\chi^2(\xi_\alpha) = \sum_i \frac{1}{\sigma_i^2} \left[ (1 + \delta^i_{\text{CS}} \xi_{\text{CS}}) \langle P_{ee} \rangle_i + \pi^i_{\theta_{12}} \xi_{\theta_{12}} + \pi^i_{\theta_{13}} \xi_{\theta_{13}} - R_i \right]^2 + \sum_{\alpha = \text{CS}, \theta_{12}, \theta_{13}} \xi_\alpha^2, \tag{13}$$

$$\pi^i_{\theta_{jk}} = \delta_{s_{jk}^2} \frac{\partial \langle P_{ee} \rangle_i}{\partial s_{jk}^2}, \qquad s_{jk}^2 \equiv \sin^2 \theta_{jk}, \quad jk = (12, 13), \tag{14}$$

where $\langle P_{ee} \rangle_i$ is the $\nu_e$ survival probability averaged over the detector volume as well as the neutrino energy lines corresponding to each data point $i$, for details see [6, 10]. As before, we take into account the asymmetric cross section uncertainties by chosing $\delta^i_{\text{CS}}$ depending on the sign of $\xi_{\text{CS}}$ at the minimum, and we include the uncertainties on the leptonic mixing angles $\theta_{12}, \theta_{13}$ by introducing the pull parameters $\xi_{\theta_{12}}, \xi_{\theta_{13}}$, and $\delta_{s_{12}^2}, \delta_{s_{13}^2}$ are the 1$\sigma$ errors on $\sin^2 \theta_{12}, \sin^2 \theta_{13}$ from NuFit-5.2 [46, 47].

The results of the fit are provided in table 3 for the two recommended cross section from eq. (10) and fitting either only the two BEST data points or all gallium data combined. Figure 1 shows the allowed parameter range for the decoherence lengths $\lambda_{12}$ and $\lambda_{13}$ using all gallium

Table 3: Best fit results for the decoherence model with $r = 2$ (left) and $r = 12$ (right) for the BEST experiment (inner and outer volumes combined) and for all gallium experiments, for the two recommended cross sections CS1 and CS2 from Haxton et al. [16]. We give the $\chi^2/$dof at the best fit point where we assume one effective fit parameter (namely $\lambda_{12}$, see text for explanations), the corresponding $p$-values of the best fit points, the $\Delta\chi^2$ to the null hypothesis, the number of two-sided Gaussian standard deviations when converting the $\Delta\chi^2$ into a confidence level for 2 dof, and the value of $\lambda_{12}$ at the best fit point. The best fit for $\lambda_{13}$ is in all cases at 0.04 m, which corresponds to the lower boundary of the considered range.

|  | $r = 2$ | | | | | $r = 12$ | | | | |
|  | $\chi^2_{\min}/$dof | $p$-val. | $\Delta\chi^2$ | $\#\sigma$ | $\lambda_{12}$ [m] | $\chi^2_{\min}/$dof | $p$-val. | $\Delta\chi^2$ | $\#\sigma$ | $\lambda_{12}$ [m] |
|---|---|---|---|---|---|---|---|---|---|---|
| CS1, BEST | 2.0/1 | 0.16 | 30.1 | 5.1 | 1.44 | 1.7/1 | 0.19 | 30.4 | 5.2 | 1.44 |
| CS1, all | 7.7/5 | 0.17 | 28.6 | 5.0 | 1.74 | 8.3/5 | 0.14 | 28.0 | 4.9 | 2.10 |
| CS2, BEST | 2.6/1 | 0.11 | 32.1 | 5.3 | 1.19 | 2.2/1 | 0.14 | 32.5 | 5.4 | 1.44 |
| CS2, all | 8.4/5 | 0.14 | 30.0 | 5.1 | 1.44 | 9.2/5 | 0.10 | 29.2 | 5.0 | 1.74 |

data and the CS2 cross section (other combinations give similar allowed regions). We consider two representative examples for the power law, namely $r = 2$ and $r = 12$. As we will see below, consistency with neutrino oscillation data requires that decoherence effects become weak very quickly as the neutrino energy increases, requiring values of $r \gtrsim 10$.

We find that the best fit point for $\lambda_{13}$ is driven towards the boundary of our considered region, at $\lambda_{13} = 0.04$ m, which effectively means full decoherence at the distances relevant for gallium experiments. In this limit the survival probability becomes

$$P_{ee}^{\text{gal}} \approx 1 - \frac{1}{2}\sin^2 2\theta_{13} - \frac{1}{2}\cos^4\theta_{13}\sin^2 2\theta_{12}\left(1 - e^{-\gamma_{12}L}\right) \qquad (\lambda_{13} \to 0), \qquad (15)$$

where we have used $\gamma_{23} \approx \gamma_{13} \gg \gamma_{12}$. Since $0.5\sin^2 2\theta_{13} \approx 0.043$, the suppression due to decoherence of the 3rd mass state is not enough to account for the $\simeq 20\%$ suppression in gallium experiments, and therefore we need to invoke decoherence in the 12 sector corresponding to the last term in eq. (15). Numerically we have $0.5\cos^4\theta_{13}\sin^2 2\theta_{12} \approx 0.404$. Hence, we need partial decoherence in the 12 sector to obtain $P_{ee} \simeq 0.8$. This is reflected in the allowed region for $\lambda_{12}$ visible in fig. 1, indicating values $\lambda_{12} \simeq 1 - 2$ m, comparable to the typical sizes of gallium experiments. From fig. 1 we also see, that the results are very similar for both sign options to determine $\lambda_{23}$ according to eq. (8), and they become identical in the limits $\lambda_{12} \gg \lambda_{13}$ and $\lambda_{12} \ll \lambda_{13}$. For definiteness we will adopt the negative sign for the following discussion.

In table 3 we provide the $\chi^2$ values at the best fit points. To calculate the corresponding $p$-value to evaluate the goodness-of-fit we assume one effective free parameter. The justification for this is that $\lambda_{13}$ is driven to small values, where predictions become independent of it, see eq. (15). This corresponds to the physical boundary $e^{-\gamma_{13}L} \leq 1$, and therefore $\lambda_{13}$ does not contribute as a full degree of freedom. This is also reflected in the result that $\chi^2_{\min}$ is non-zero when fitting only the two BEST data points, indicating that the number of effective degrees of freedom is less than 2. In all cases shown in the table we find $p$-values in the range between 10% and 20%. While this is a huge improvement compared to the $p$-values of the null hypothesis (see table 2) the fit is not perfect. This is related to the partial decoherence in the 12 sector, which is required for the reasons discussed above. It leads to a distance dependence on the scale of gallium experiments which in particular predicts different event ratios in the inner and outer detector volumes of the BEST experiment. We illustrate this on one example fit in fig. 2 which compares the predicted ratios at the best fit point to the observed values. While currently this is acceptable within uncertainties, the distance dependence of $P_{ee}$ at the

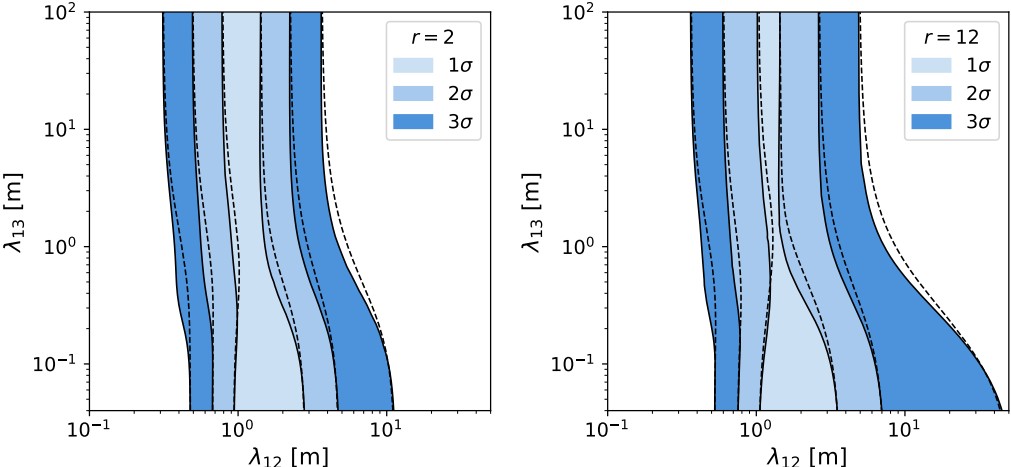

Figure 1: Allowed regions for the decoherence lengths $\lambda_{12}$ and $\lambda_{13}$ at $1, 2, 3\sigma$ for 2 dof obtained by fitting combined gallium data. The left (right) panel corresponds to an energy dependence of the decoherence parameter with the power $r = 2$ (12). We use the CS2 cross section. The black-solid contours/blue regions assume $\gamma_{23} = \gamma_{12} + \gamma_{13} - 2\sqrt{\gamma_{12}\gamma_{13}}$ whereas the dashed contours use $\gamma_{23} = \gamma_{12} + \gamma_{13} + 2\sqrt{\gamma_{12}\gamma_{13}}$.

scale of 1 m and few 100 keV neutrino energies is a specific prediction of this scenario.

In table 3 we also provide the $\Delta\chi^2$ of the best fit points with respect to the null hypothesis. Here we use 2 dof to evaluate these values as both parameters, $\lambda_{12}$ and $\lambda_{13}$, have to be changed to move from the best fit point to the null hypothesis which corresponds to $\lambda_{12,13} \to \infty$. We obtain that the decoherence model is preferred over the null hypothesis at the level of around $5\sigma$ in all cases considered in the table. The actual distribution of the $\Delta\chi^2$ needs to be determined by MC simulations, to account for deviations from Wilks' theorem. For similar arguments as given above, one may expect that the true number of degrees of freedom is between 1 and 2. This can be understood by noting that $0 \leq e^{-\gamma_{ij}L} \leq 1$ when changing $\lambda_{ij}$ from zero to infinity. The coefficient in front of $e^{-\gamma_{13}L}$ is small ($0.5\sin^2 2\theta_{13} \approx 0.043$), and therefore the variation in the prediction is limited to a small range, even when $\lambda_{13}$ is varied from zero to infinity. Therefore, it does not contribute as a full degree of freedom to the fit. This is different for $\lambda_{12}$, which allows for large variations in the predictions and therefore contributes as a full degree of freedom. For 1 dof, the significance in number of standard deviations is given by $\sqrt{\Delta\chi^2}$. In table 3 we have decided to use instead the conservative choice of 2 dof, which leads to slightly lower significances; we expect the "correct" answer to lie in-between the significanes for 1 and 2 dof.

We note that decoherence in the 13 sector is actually not required by the fit; the allowed

Table 4: Same as table 3 but setting $\lambda_{13} \to \infty$. The number of standard deviations relative to the null hypothesis are obtained by evaluating $\Delta\chi^2$ for 1 dof.

| | $r = 2$ | | | | | $r = 12$ | | | | |
|---|---|---|---|---|---|---|---|---|---|---|
| | $\chi^2_{\min}$/dof | $p$-val. | $\Delta\chi^2$ | #$\sigma$ | $\lambda_{12}$ [m] | $\chi^2_{\min}$/dof | $p$-val. | $\Delta\chi^2$ | #$\sigma$ | $\lambda_{12}$ [m] |
| CS1, BEST | 3.0/1 | 0.08 | 29.1 | 5.4 | 0.99 | 2.6/1 | 0.11 | 29.5 | 5.4 | 1.12 |
| CS1, all | 9.1/5 | 0.10 | 27.2 | 5.2 | 1.27 | 10.3/5 | 0.07 | 26.0 | 5.1 | 1.44 |
| CS2, BEST | 3.5/1 | 0.06 | 31.2 | 5.6 | 0.87 | 3.1/1 | 0.08 | 31.6 | 5.6 | 0.93 |
| CS2, all | 9.8/5 | 0.08 | 28.6 | 5.4 | 1.05 | 10.3/5 | 0.07 | 28.1 | 5.3 | 1.44 |

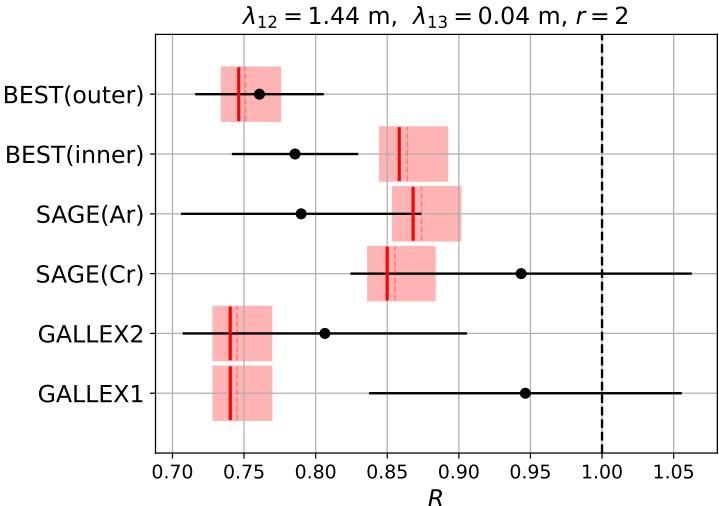

Figure 2: Predicted event ratios at the best fit point for the combined gallium data for $r = 2$ and the CS2 cross section (red lines). The red shaded boxes indicate the $1\sigma$ correlated cross section uncertainty on the predictions. Black data points show the observed ratios with error bars at $1\sigma$ including statistical and experimental systematic errors.

regions at $1\sigma$ extend up to $\lambda_{13} \to \infty$, c.f. fig. 1. In table 4 we give the properties of the best fit results when fixing $\lambda_{13} \to \infty$, i.e., $\gamma_{13} = 0$. In this case, we have $\lambda_{12} = \lambda_{23}$ and the survival probability relevant for gallium data becomes[1]

$$P_{ee}^{\text{gal}} \approx 1 - 2|U_{e2}|^2(1 - |U_{e2}|^2)\left(1 - e^{-\gamma_{12}L}\right) \qquad (\lambda_{13} \to \infty). \qquad (16)$$

Comparing the $\chi^2_{\min}$ values from tables 3 and 4, we see that the $\chi^2$ is increased only by about 1 unit, the goodness-of-fit is around or slightly below 10% for all cases, whereas the preference compared to the null hypothesis is above $5\sigma$ in all cases.

## 4 Consistency with oscillation data

In this section we show that our proposed explanation of the gallium anomaly does not impact the various observations of neutrino oscillations. We focus first on solar neutrinos in section 4.1, which require special care due to the matter effect in the sun. This will lead us to a very steep energy dependence of the decoherence coefficients with $r \gtrsim 10$. The remaining oscillation data is discussed in section 4.2 where we also consider short-baseline anomalies and argue that in certain decoherence scenarios the LSND anomaly could be explained as well. The relevant length scales are illustrated in fig. 3.

### 4.1 Solar neutrinos

As discussed in section 2, if matter effects can be neglected, the decoherence terms lead to a damping of the off-diagonal elements of the density matrix in the mass basis, which results in an incoherent mixture of neutrino mass eigenstates. In other words, they suppress the interference terms of the oscillation probability, see eq. (5). For distances larger than the oscillation

---

[1]An equivalent solution is obtained for $\lambda_{12} = \lambda_{13}$ with $\lambda_{23} \to \infty$. This case corresponds to replacing $|U_{e2}|^2 \to |U_{e1}|^2$ in eq. (16).

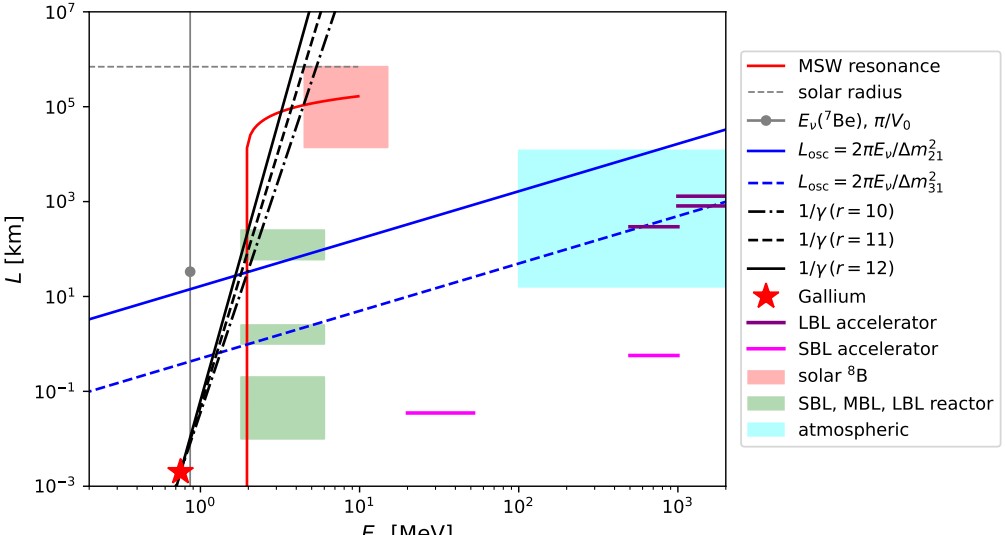

Figure 3: Comparison of relevant length scales at different neutrino energies. Lines in black show the decoherence length $1/\gamma$ for $\lambda = 2$ m and $r = 10, 11, 12$; the region around and above these lines is affected by the decoherence terms. Blue lines show the vacuum oscillation lengths due to $\Delta m_{21}^2$ and $\Delta m_{31}^2$. Furthermore, we show approximately the regions probed by gallium experiments (red star), short-, medium-, and long-baseline reactor experiments (green regions), atmospheric neutrinos (cyan region), as well as accelerator experiments including the long-baseline experiments T2K, NOvA, DUNE (purple) and short-baseline experiments LSND and MiniBooNE (magenta). The red curve shows the distance of the MSW resonance inside the sun from the solar center. We also indicate the energy of the $^7$Be solar neutrino line and the size of the matter potential at the center of the sun converted into a distance (grey), as well as the energy range relevant for $^8$B solar neutrinos (red region).

length, these terms average to zero even for standard evolution with $\gamma_{ij} = 0$. This means that decoherence does not change the oscillation probability in such a case. In other words, for the range well above the blue solid line in fig. 3, the oscillatory terms in the oscillation probability average to zero and decoherence effects cannot in practice be resolved, regardless of whether we are above the black lines or not. As a result, the effect of decoherence on the oscillation of solar neutrinos from the Sun surface to the Earth surface (as well as for the supernova or cosmic neutrinos outside the source) will not be observable.

However, inside the Sun and the Earth, the matter effects [48, 49] change the picture [25, 27, 30, 31]. While $D$ commutes with the Hamiltonian in vacuum, it will not commute with the effective Hamiltonian inside matter. In such a situation the decoherence terms will push $\rho$ towards a matrix proportional to the identity matrix. This can be understood because in this limit both $\mathcal{D}$ and the commutator of $\rho$ and the Hamiltonian vanish, making $\rho \propto I$ an asymptotic solution of eq. (1) when $[D_n, H] \neq 0$. Let us discuss the various parts of the solar neutrino spectrum in turn.

**Low energy:** Solar $pp$ neutrinos with energies $\lesssim 0.4$ MeV will be strongly affected by decoherence. However, for neutrinos with these energies, matter effects are small and their survival probability is determined by vacuum oscillations. As mentioned above, in this case, the decoherence effects are indistinguishable from standard averaging and hence we expect no modification of low energy solar neutrinos compared to the standard oscillation picture.

**High energy:** Let us now focus on $^8$B neutrinos with energies above the SK detection threshold of 4.5 MeV [50]. The relevant region is indicated by the red-shaded box in fig. 3.[2] When the high-energy solar neutrinos propagate out from the center of the sun to the surface, the evolution follows adiabatically the effective mass eigenstates in matter until they cross the MSW resonance. After the resonance we have basically propagation of the vacuum mass states. The red curve in fig. 3 shows the location of the MSW resonance in the Sun as a function of neutrino energy. Below 2 MeV, the density even in the Sun center will be too low for a resonance. In order to be consistent with the success of the MSW mechanism we need to make sure, that the decoherence effects do not affect the evolution as long as matter dominates. Hence, we require that the decoherence length $1/\gamma$ must be larger than the path-length during which matter dominates. We can see from the figure, for a decoherence length as required to explain gallium data at $E_\nu \simeq 0.75$ MeV, we need $r \gtrsim 10$ to have $1/\gamma$ larger than the resonance location for $E_\nu > 4.5$ MeV. We have verified also by numerical calculations, that for these values, the decoherence effects do not modify significantly the $\nu_e$ survival probability in this energy range. Note that this requirement ensures also that the day-night effect for the $^8$B solar neutrinos will not be modified compared to the standard theory, as the decoherence length is many orders of magnitudes longer than the Earth diameter.

**Intermediate energy:** As indicated by the grey line in fig. 3, at $E_\nu = 0.862$ MeV, which is the energy of $^7$Be line measured by Borexino [51, 52], $\gamma$ can be sizable. Since this energy is close to the one in gallium experiments, $\gamma$ for the $^7$Be line is not significantly suppressed relative to that at the gallium experiments. However, in this case, the matter effects are subdominant: $[\Delta m_{21}^2/(2E_\nu)]/(\sqrt{2}G_F n_e|_{\text{Sun center}}) \sim 0.1$, and we expect the decoherence effects to be approximately similar to the pure vacuum case. Numerical computation shows that the deviation of $P_{ee}$ from the standard prediction is at the level of 10% which is of the same size as the current experimental uncertainty at $1\sigma$ and therefore compatible with observations: $P_{ee}(0.862\,\text{MeV}) = 0.53 \pm 0.05$ [52].

In summary, we conclude that by choosing $r \gtrsim 10$ we can make our gallium explanation consistent with solar neutrino data. Notice that we cannot avoid this solar neutrino bound on $r$ by taking different decoherence parameters for neutrinos and antineutrinos. In future, with precise measurements of the vacuum-to-matter transition region of the solar $\nu_e$ survival probability we may be able to detect deviations from the standard MSW prediction. We leave a dedicated investigation of this possibility for future work.

## 4.2 Other oscillation data

We now comment on the impact of decoherence on other neutrino oscillation experiments. We note that values of $r \neq 1$ imply violation of Lorentz symmetry. Therefore, it may be expected that decoherence effects are also CPT violating and $\gamma_{ij}$ could be different for neutrinos and antineutrinos [23]. In such a case, we make no prediction for parameters for antineutrinos (see also the discussion of LSND below on this point).

If we assume that decoherence is the same for neutrinos and antineutrinos, reactor experiments put also a sever constraint on the energy dependence of $\gamma_{ij}$. In fig. 3 the green boxes indicate the ranges probed by various classes of reactor experiments: short-baseline experiments at $L \lesssim 100$ m, medium-baseline experiments, such as DayaBay, RENO, DoubleChooz at $L \sim 1$ km, and the long-baseline experiment KamLAND with $L \sim 180$ km. The spectral distortion observed in the latter [53] poses a sever constraint on decoherence effects (see e.g., [25, 54] for related analyses). Figure 3 suggests, that values $r \gtrsim 12$ are required for not

---

[2]The $L$ range for this box is only for illustration purposes and has been chosen as $[0.02R_\odot, R_\odot]$, with $R_\odot$ denoting the solar radius and $0.02R_\odot$ is approximately the production region for $^8$B neutrinos inside the Sun.

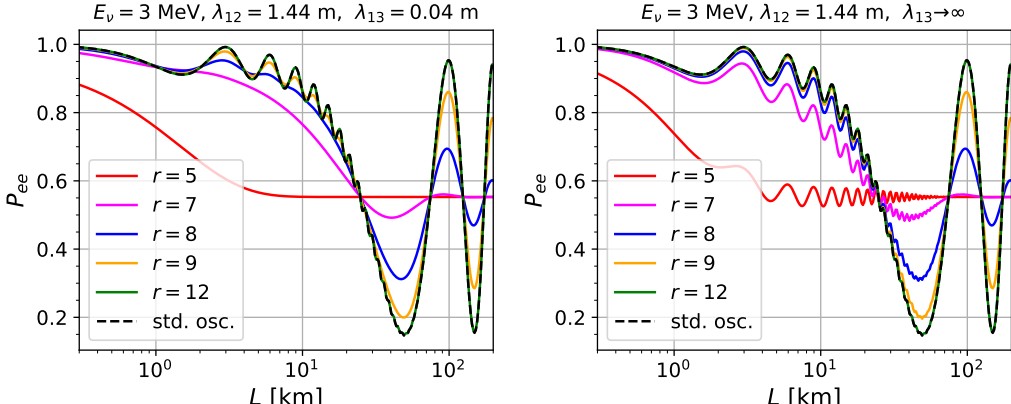

Figure 4: Survival probability $P_{ee}$ as a function of the baseline $L$ for $E_\nu = 3$ MeV with the decoherence lengths $\lambda_{12} = 1.44$ m (both panels) and $\lambda_{13} = 0.04$ m ($\infty$) for the left (right) panel and for several values of $r$. The black dashed curve corresponds to the standard three flavour oscillation probability, which overlaps with the $r = 12$ curve. Oscillation parameters are taken at the NuFit-5.2 best fit point [47]. Probabilities are averaged over a Gaussian energy resolution of $0.03\%\sqrt{\text{MeV}/E_\nu}$.

affecting KamLAND. Figure 4 shows the survival probability relevant for reactor experiments as a function of distance, assuming that neutrinos and antineutrinos are subject to the same decoherence effects. We see that in order to be consistent with KamLAND we need a very steep energy dependence, $r \gtrsim 10$, in order to compensate the factor $L_{\text{KamL}}/L_{\text{Gal}} \sim 200\,\text{km}/(2\,\text{m}) = 10^5$ by the factor $(0.75\,\text{MeV}/E_\nu)^r$. The future JUNO reactor experiment at $L \simeq 60$ km may be able to further strengthen the requirement on $r$.

From fig. 3 it is clear that for all the other oscillation experiments, including atmospheric and accelerator neutrino experiments, decoherence effects on our model will be negligible, if the power law extends to $E_\nu \gtrsim 0.1$ GeV.

**LSND, MiniBooNE and short-baseline reactors.** From figs. 3 and 4 it becomes clear, that in our scenario short-baseline reactor experiments are not affected: decoherence effects at short baselines would spoil the oscillation signatures observed at medium and long-baseline reactor experiments. Similarly, we cannot explain the MiniBooNE anomaly [35], see magenta bar around $10^3$ MeV in fig. 3: decoherence at such small baselines would distort the oscillation signatures observed in atmospheric and long-baseline accelerator experiments. In both cases (short-baseline reactor experiments and MiniBooNE), decoherence effects are completely negligible under the power law assumption with $r \gtrsim 10$.

The LSND experiment, reporting evidence for $\bar{\nu}_\mu \to \bar{\nu}_e$ transitions [34], corresponds to the magenta bar around 30 MeV in fig. 3. If we assume the same decoherence parameters for neutrinos and antineutrinos and the power law with $r \gtrsim 10$, it is clear that no effect is predicted for LSND. However, as there are no other observations in this energy range,[3] we can introduce decoherence effects there to explain LSND as well, for instance adopting a scenario as in ref. [28]. This could be achieved in the following two ways:

- We could assume that the decoherence effects violate the CPT symmetry and postulate different decoherence parameters for neutrinos and antineutrinos. To explain the

---

[3]Note that within the standard model, coherent neutrino–nucleus scattering as observed by COHERENT [55] is a flavour-universal neutral-current process and is therefore not expected to be affected by flavour transitions due to decoherence. However, in the presence of decoherence, the bounds on new physics such as non-standard neutrino interactions with non-universal couplings should be reconsidered.

gallium anomaly we need decoherence in neutrinos at $E_\nu \simeq 0.75$ MeV getting quickly suppressed for higher energies. For LSND we need decoherence around $E_\nu \simeq 30$ MeV in antineutrinos, being suppressed both at lower and at higher energies [28].

- To explain both gallium and LSND anomalies with the same parameters for neutrinos and antineutrinos, we may consider a double-peak structure of the decoherence effects, occurring both at $E_\nu \simeq 0.75$ MeV and around 30 MeV, but being strongly suppressed in between and above these energies. For example, this can be achieved by setting $d_1 = 0$, $d_2$ a Gaussian peaked around 0.75 MeV and $d_3$ another Gaussian peaked around 30 MeV (see eq. (4) for the relation between $d_i$ and the decoherence prameters $\gamma_{ij}$).

To identify possible UV completions for such scenarios is beyond the scope of the present article.

## 5 Summary

We have proposed an explanation of the gallium anomaly based on quantum decoherence. Our scenario does not require sterile neutrinos, but we postulate that at the relevant neutrino energies of $E_\nu^{\text{gallium}} \simeq 0.75$ MeV the neutrino mass states $\nu_1$ and $\nu_2$ decohere at length scales comparable to the size of the gallium detectors, of order 2 m. In order to be consistent with other oscillation data, in particular with solar neutrinos and (if equal decoherence parameters for neutrinos and antineutrinos are assumed) the KamLAND reactor experiment, decoherence effects have to decrease quickly for energies larger than $E_\nu^{\text{gallium}}$. If we assume a power law behaviour with neutrino energy, the decoherence parameters should scale as $E_\nu^{-r}$ with $r \gtrsim 10-12$. While explanations of the gallium anomaly in terms of sterile neutrinos with eV-scale mass-squared differences suffer from severe tension with solar neutrinos and reactor data, the explanation proposed here is consistent with these data. Furthermore, we expect that cosmology is not changed compared to the standard three-flavour neutrino case.

With the power law energy dependence mentioned above, all other data on neutrino oscillations will be unaffected and proceed as in the standard three-flavour scenario, including experiments at short baselines. However, it may be possible to reconcile our proposal for gallium also with an explanation of LSND in terms of decoherence, if we allow for different decoherence parameters for neutrinos and antineutrinos or by adopting a peaked energy dependence for the decoherence parameters.

A testable prediction of our scenario is a distance dependent deficit at the radioactive source experiments. We predict a $\nu_e$ survival probability of about 0.86 in the inner detector volume of the BEST experiment and 0.75 in the outer volume. Although the current BEST results do not show evidence for such a behaviour, our prediction is consistent with the BEST data within the errors. If more precise measurements in the future confirm a distance-independent suppression at the scale of 1–2 meters, our proposal can be ruled out. Another signature of this model is the modification of the solar neutrino survival probability in the transition region between the vacuum and matter dominated energy regimes. The possibility to test this model by future high-precision solar neutrino observations requires further study.

## Acknowledgments

We thank Peter Denton for asking a relevant question related to the statistical analysis, which helped us to improve the paper.

**Funding information** This work has been supported by the European Union's Framework Programme for Research and Innovation Horizon 2020 under grant H2020-MSCA-ITN-2019/860881-HIDDeN. YF has received financial support from Saramadan under contract No. ISEF/M/401439. She would like to acknowledge support from ICTP through the Associates Programme and from the Simons Foundation through grant number 284558FY19 as well as under the Marie Skłodowska-Curie Staff Exchange grant agreement No 101086085 - ASYMMETRY.

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
