# Peer review of "A decoherence explanation of the gallium neutrino anomaly"

_SciPost Physics, doi:SciPost Phys. 15, 172 (2023)_

## Round 1 · Referee Report · Anonymous (Referee 1) · 2023-9-6

Report
This paper presents a decoherence explanation of the gallium neutrino anomaly.
It is assumed that decoherence happens for the low-energy neutrinos in gallium experiments but not for the higher energy neutrinos in other experiments, because of a very steep dependence on energy.
I think that this explanation of the gallium neutrino anomaly is rather ad hoc, but it cannot be excluded.
It is however interesting, because it does not need a sterile neutrino or other non-standard effect beyond the decoherence.
It is also interesting that it can be tested in the future by checking the prediction of a distance-dependent deficit for the radioactive source experiments.
The paper is clear and well-written.
In conclusion, I recommend the publication of this paper.
It is assumed that decoherence happens for the low-energy neutrinos in gallium experiments but not for the higher energy neutrinos in other experiments, because of a very steep dependence on energy.
I think that this explanation of the gallium neutrino anomaly is rather ad hoc, but it cannot be excluded.
It is however interesting, because it does not need a sterile neutrino or other non-standard effect beyond the decoherence.
It is also interesting that it can be tested in the future by checking the prediction of a distance-dependent deficit for the radioactive source experiments.
The paper is clear and well-written.
In conclusion, I recommend the publication of this paper.

---

## Round 1 · Referee Report · Anonymous (Referee 2) · 2023-9-10

Strengths
Originality
Clear presentation
Possibility of testing the proposed explanation
Clear presentation
Possibility of testing the proposed explanation
Report
This paper discusses an explanation of the gallium neutrino anomaly
based on decoherence effects. Due to the peculiar energy dependence of the decoherence effect, only low-energy neutrinos relevant to the Gallium experiments are affected, while experiments involving high-energy neutrinos are essentially unmodified. Even if similar decoherence effects have not been observed so far, such an explanation is very economical since it does not require enlarging the particle content of the Standard Model. Future experiments will be able to test the explanation.
I recommend the publication of this paper in its present form.
based on decoherence effects. Due to the peculiar energy dependence of the decoherence effect, only low-energy neutrinos relevant to the Gallium experiments are affected, while experiments involving high-energy neutrinos are essentially unmodified. Even if similar decoherence effects have not been observed so far, such an explanation is very economical since it does not require enlarging the particle content of the Standard Model. Future experiments will be able to test the explanation.
I recommend the publication of this paper in its present form.

---

## Editorial Decision

published